# Car Ownership Behavior Model Considering Nonlinear Impacts of Multi-Scale Built Environment Characteristics

**Lan Wu** [1,*], **Xiaorui Yuan** [1], **Chaoyin Yin** [1], **Ming Yang** [2] **and Hongjian Ouyang** [1]

1   College of Automobile and Traffic Engineering, Nanjing Forestry University, Nanjing 210037, China; cyyin@njfu.edu.cn (C.Y.)
2   Nanjing Institute of City and Transport Planning Co., Ltd., Nanjing 210008, China
*   Correspondence: wulan@njfu.edu.cn

**Abstract:** To explore the nonlinear influence of a multi-scale built environment on residents' car ownership behavior, combined with the data set of residents' individual information and travel-related data from the China Labor Force Dynamic Survey report, eight variables are selected to describe the built environment from multiple scales. The gradient-boosting iterative decision tree model including individual family attributes and neighborhood-scale and city-scale built-environment attributes is constructed. The results show that the individual family attributes have the greatest cumulative impact on car ownership behavior (46.3%). The built environment based on neighborhood scale and city scale also has a significant impact on residents' car ownership behavior, these being 33.94% and 19.76%, respectively. The distance to the city center at the neighborhood scale is positive correlated with car ownership. The number of buses per 10,000 people and road area per capita in the city scale are also positive correlated with car ownership. Therefore, in order to slow down the increase in car ownership, the built environment can be optimized and adjusted at neighborhood scale and city scale.

**Keywords:** urban traffic; built environment; gradient boosting decision tree; car ownership; nonlinear effects

## 1. Introduction

With the continuous acceleration of urbanization and the increasing improvement of people's living standards, residents' daily travel has gradually deepened their dependence on cars. The resulting environmental pollution, road congestion and waste of resources have become one of the problems restricting the sustainable development of major cities [1]. In order to alleviate the problems of natural resources, health and environment caused by the increasing dependence of residents on cars, this paper starts with the factors that affect the behavior of residents' car ownership, and uses urban traffic planning and related policy measures to guide residents' green travel consciousness [2–4], thereby reducing residents' dependence on cars, which has become a hot issue in the field of urban traffic planning research.

The urban built environment determines the spatial distribution of residents' daily activities and has an important impact on individual travel behavior [5–7]. Scholars have revealed the impact of the built environment on car ownership from different perspectives. For the overall urban built environment, Zegras et al. [8] constructed a vehicle selection model to explore the relationship between the built environment and household car ownership, and found that the built environment can have a significant impact on residents' car ownership behavior. Ding et al. [9] used the multi-layer hybrid ordered Probit model to capture the spatial heterogeneity of the traffic analysis area to evaluate the impact of the built environment on the level of household car ownership. The results show that the model can effectively solve the problem of spatial heterogeneity between regions, and has better model fitting rows than the traditional ordered Probit model. Yang et al. [10] used Nanjing

residents' travel survey data for modeling research. The study found that the built environment has a significant impact on household car ownership. The government can guide residents' travel habits by improving the urban structure. Guan et al. [11] believed that individual travel behavior stems from the distribution of family tasks and resources (such as family vehicles). The travel behavior of family members is interdependent. Through research, it was found that the built environment can indirectly affect the travel behavior of other members by affecting the travel choice of family members. Using exploratory factor analysis and multiple linear regression, Anirudh et al. [12] used the large data set of the 2011 Socio-Economic Caste Census (SECC) (including 2075 cities) to find that the regional background of the city has a significant impact on urban car ownership.

The measurement methods of the destination-accessibility dimension in existing research are mainly the distance to the central business district (CBD), the distance to the city center and the distance to the city sub-center [13,14]. Most studies have focused on the impact of CBD distance on car ownership. Since the distance to CBD is an important representation of the spatial location of the built environment in the community layer, most research results show that the distance to CBD is an important variable affecting car ownership [15]. With further in-depth research, some scholars have found that in multi-center cities, the distance to the urban sub-center is also an important variable that affects car ownership behavior. Shen et al. [16] analyzed the impact of the built environment on car ownership behavior by modeling residents' travel data in Shanghai. The results show that the distance to CBD has a significant impact on car ownership.

The measurement methods of the public-transport-accessibility dimension in existing research are mainly the distance to public transport stations and the density of public transport stations [17,18]. Studies have shown that there is a certain relationship between urban public transport construction and residents' car ownership [19]. Taking Guangzhou as an example, Huang et al. [20] selected the number of bus stops within 500 m of a residence, the distance to the nearest public transport station, the density of public transport networks within 500 m of the residence, the shortest travel time by bus and the distance to the nearest subway station to characterize the accessibility of public transport, and modeled and analyzed the car ownership behavior of residents. Ding et al. [21] employed a gradient-boosting machine (GBM) based on the relationship between the attributes of the built environment and residents' traffic commuting to explore the nonlinear relationship. The research shows that the density of bus stations and the distance from bus stations have a greater impact on residents' travel modes. In addition, Ding et al. [22] constructed a semi-parametric multi-level mixed Logit model to study the nonlinear and spatial heterogeneous relationship between built-environment attributes and public transportation. The results show that there is a significant nonlinear correlation between the built-environment attributes of residential communities and transit commuting behavior. Chen et al. [23] found that land use mix has a great impact on residents' choice of travel mode, and through the analysis of urban land use types, it was found that the impact of non-commuting activities on travel behavior in large cities was expanding compared with commuting activities.

In terms of the measurement of built-environment diversity, the existing research mostly uses land use hybridity as an indicator, because this indicator can reflect the target accessibility on a regional scale, and the higher the target accessibility, the greater the possibility that residents would choose non-motorized travel [24,25]. Yin et al. [26] quantitatively measured the degree of urban road-traffic mixing by constructing correlation indicators, and found through empirical research that the degree of urban road-traffic mixing has a significant negative impact on urban road-traffic travel patterns. In addition, Sarkar et al. [27] established a multivariate Logit model based on the traffic survey data of Agartala City, and analyzed the urban traffic mode and traffic distance. The results show that increasing the degree of urban traffic mixing can not only reduce urban traffic, but also reduce the probability of using urban transportation. Zhang et al. [28] used the Gradient-Boosting Decision Tree (GBDT) model to study the relative importance of regional accessibility and land mix to reduce residents' driving distance, and found that multi-center

development and neighborhood life-circle planning would reduce residents' vehicle utilization. Shirgaokar [29] analyzed the influence of the residential building environment on vehicle mileage by a generalized linear model. The results of the model indicate that the combination of land use has a significant negative effect on the mileage of vehicles. Based on two rounds of survey data in China, Yin et al. [30] found that promoting more balanced land use and increasing residential density are very important for reducing car use and active travel activities.

By using the polynomial structural equation model, Yin discussed the relationship between the built environment and the mileage of residents' vehicles [31]. The results show that the built environment has a significant effect on the ownership and use of cars. Similarly, several articles have analyzed the relationship between urban transportation and car ownership and use, and found that built environments can affect residents' dependence on cars [32]. Some scholars have also studied the neighborhood-scale built environment.

Recently, some scholars have analyzed the impact of the built environment on household car ownership based on a hierarchical Bayesian model, and found that the built environment on a neighborhood scale has a significant impact on car ownership decision-making behavior [33]. Yin et al. [34] found that the built environment of a residence is an important factor affecting residents' car ownership behavior in their study of parking availability in the residence and workplace. Subsequently, some scholars began to study the impact of the multi-scale built environment on residents' car ownership. Doddamani et al. [35] explored the relationship between the built environment and household car ownership using survey data from the two cities of Hubli and Dharwad in India, and found that among built-environment variables, land mix and population density can reduce urban car ownership. Yin et al. [36] used a multi-level Logit regression model to regress the built environment and individual socioeconomic attributes on car ownership at the urban and neighborhood scales. The study found that the built environment at neighborhood and city scales significantly affects car ownership. Zhang et al. [37] used the gradient-boosting decision tree algorithm to study the impact of accessibility on household car ownership. The results show that local accessibility, such as the density of places of retail and service types and the density of workplaces, is more important to car ownership. Ao et al. [38] constructed a multinomial logit model to estimate the number of car-owning residents, and found that family attributes and built-environment attributes were significantly related to residents' car ownership. Wang et al. [39] modeled residents' car ownership behavior from the two levels of workplace and residence, and found that the built-environment attributes of both places have a significant impact on residents' car ownership behavior.

The existing research has focused on the impact of the built-environment characteristics on car ownership, usually using statistical models to analyze the linear relationship between variables, ignoring the nonlinear relationship between them. However, further studies have found a significant nonlinear relationship between built-environment characteristics and travel behavior. Therefore, it is necessary to construct a machine-learning model for car ownership behavior in different scales of built environment to explore the nonlinear relationship between them. Figure 1 is the research framework of this paper.

On the basis of existing research, this paper considers the individual family attributes at the individual scale, the built-environment attributes at the neighborhood scale, and the built-environment attributes at the city scale. The Gradient-Boosting Decision Tree (GBDT) is used to construct a model of residents' car ownership against the nonlinear effect of the urban multi-scale built environment, to quantify the impact of various attributes on residents' car ownership behavior, explore the nonlinear effect of the multi-scale built environment on car ownership behavior, and provide a theoretical basis for precise urban planning and transportation policy formulation.

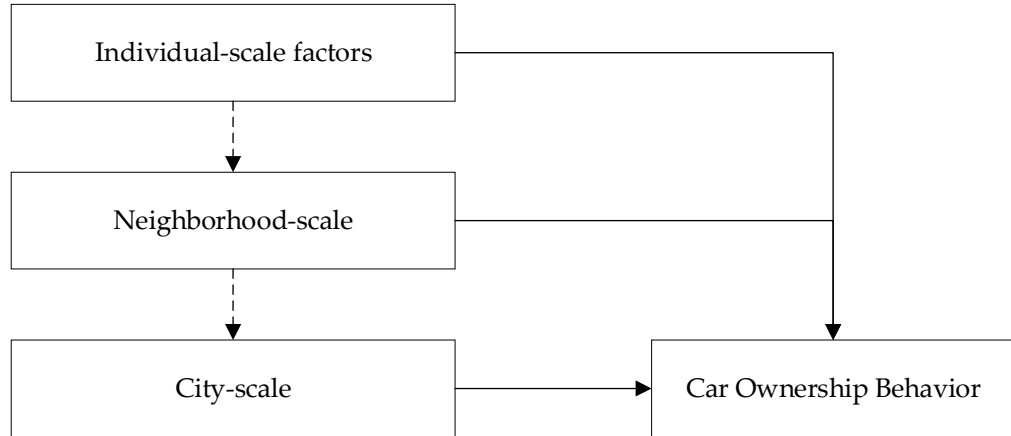

**Figure 1.** Research Framework Diagram.

## 2. Research Data

The data mainly comes from the comprehensive report on the labor force dynamic survey organized by the social science survey center of Sun Yat-sen University in 2014. After screening and cleaning the initial data, a total of 17,186 valid samples were obtained, covering 116 cities across the country, a total of 334 communities, and the rest of the data were obtained from the statistical yearbook reports of each city. Table 1 is the descriptive statistical results of residents' individual family attributes, neighborhood and the built-environment characteristics of the city based on geographical location.

**Table 1.** Statistical description of each variable.

| Variable | Description | Mean | SD |
|---|---|---|---|
| **Individual level—personal family attributes** | | | |
| Gender | 1: male; 0: female | 0.48 | 0.5 |
| Age | Age of respondents | 40.92 | 12.17 |
| Education | 1: Primary; 2: Middle; 3: High; 4: University and above | 2.12 | 1.03 |
| Family size | Number of respondents' family members | 4.50 | 1.96 |
| Number of members under 16 years old | Number of family members under 16 years old | 0.47 | 0.5 |
| Household Income (yearly) | Family annual income (10,000 RMB) | 6.38 | 13.14 |
| **Neighborhood scale—built environment characteristics** | | | |
| Degree of mixed land use | Calculated based on 5 facilities: sports squares, libraries, hospital clinics, schools and banks. | 0.61 | 0.3 |
| Distance to bus station | Walking distance to the nearest bus station (km) | 2.21 | 5.27 |
| Distance to city center | Distance from residence to the center of city (km) | 4.79 | 6.79 |
| Neighborhood Population density | Ratio of neighborhood population to neighborhood area (10,000 persons/km$^2$) | 1.11 | 5.44 |
| **City scale—built environment characteristics** | | | |
| City population density | Ratio of urban resident population to urban built-up area (10,000 persons/km$^2$) | 0.21 | 0.3 |
| Whether there is a subway | 1: Yes; 0: No | 0.26 | 0.44 |
| The number of buses per 10,000 people | Number of buses per 10,000 people (Vehicles) | 10.89 | 11.77 |
| Road area per capita | Ratio of urban road area to urban resident population (m$^2$/person) | 16.85 | 9.03 |

The built environment is described from the two levels of community and city. The built environment of the community layer is derived from the community and family data files of the residents, and the relevant data are directly derived or calculated from the survey data set. The basic data of the built environment of the community layer are derived from the government's documentary report, and the specific traffic distance is

from the household file of the data report. The urban built environment is derived from the statistical yearbook report. The specific variable data are directly derived or calculated from the city's statistical yearbook report.

The characteristics of a city's built environment are measured from two spatial scales: neighborhood scale and city scale. The built-environment characteristics at the neighborhood scale include four variables: the degree of mixed land use, distance to the bus station, distance to the city center and neighborhood population density. The distance to the bus station characterizes the accessibility of the built environment at the neighborhood scale; the distance to the city center characterizes the convenience of the built environment at the neighborhood scale; neighborhood population density is obtained by the ratio of the population in the neighborhood to the administrative area of the neighborhood, which is used to characterize the population density of the built environment at the neighborhood scale; the land-use mix is calculated based on five types of facilities in the neighborhood and surrounding area (sports squares, libraries, hospital clinics, schools and banks), and is used to characterize the 'diversity' of the built environment at the neighborhood scale. The specific value of the degree of mixed land use is obtained by the entropy index method. The formula is as follows:

$$M = \frac{-\sum p_{ij} \ln p_{ij}}{\ln N_j} \tag{1}$$

where $p_{ij}$ is the proportion $i$ of facilities in the neighborhood $j$; $N_j$ is the total number of facilities in the neighborhood $j$.

The characteristics of the city-scale built environment are represented by four attributes: urban population density, whether there is a subway, the number of buses per 10,000 people and road area per capita. The density of the urban resident population is obtained from the ratio of the total urban resident population to the area of the urban built-up area, which is used to represent the population distribution of the city; the number of buses per 10,000 people and the subway is used to represent public transport development; the road area per capita is used to characterize the degree of road construction of the city. The data of urban built-environment characteristics are from the statistical yearbook report of the corresponding city.

In order to avoid a serious multicollinearity problem between the selected independent variables, which would lead to the deviation of the final results of the model, it was necessary to test the multicollinearity of the selected independent variables before conducting the modeling research. The variance inflation factor (VIF) method was selected to test the multicollinearity of the model. The calculation formula is shown in Equation (2).

$$VIF = \frac{1}{1 - R_i^2} \tag{2}$$

where $R_i^2$ is the multiple correlation coefficient of the independent variable $x_i$ for the regression analysis of the remaining $p - 1$ independent variables.

$R_i^2$ represents the coefficient of determination in linear regression. The closer $R_i^2$ is to 1, the stronger the correlation between the variables, that is, the more severe the multicollinearity between the variables, the greater the variance expansion coefficient. When the VIF value is greater than 10, it indicates that there is serious multicollinearity between variables. The dependent variable selected in this paper is whether the residents have cars. The multicollinearity between the independent variables, including the individual family attributes of the individual layer, the neighborhood scale and the built-environment characteristics at the city scale, was tested by STATA software; the test results are shown in Table 2. According to the test results, the independent variables selected all passed the multicollinearity test, which shows that the independent variables selected in the subsequent modeling research were available.

**Table 2.** Results of variable multicollinearity test.

| Variable | VIF | Tolerance |
|---|---|---|
| **Individual level—personal family attributes** | | |
| Gender | 1.03 | 0.97 |
| Age | 1.45 | 0.69 |
| Education | 1.43 | 0.70 |
| Family size | 1.58 | 0.63 |
| Number of members under 16 years old | 1.58 | 0.63 |
| Income(yearly) | 1.04 | 0.96 |
| **Neighborhood scale—built environment characteristics** | | |
| Degree of mixed land use | 1.03 | 0.97 |
| Distance to bus station | 1.11 | 0.90 |
| Distance to city center | 1.1 | 0.91 |
| Neighborhood population density | 1.02 | 0.98 |
| **City scale-built environment characteristics** | | |
| City population density | 1.06 | 0.94 |
| Whether there is a subway | 1.36 | 0.74 |
| The number of buses per ten thousand people | 1.41 | 0.71 |
| Road area per capita | 1.18 | 0.85 |

## 3. Model Construction

The GBDT model is constructed with individual family attributes, neighborhood-scale and city-scale built-environment attributes as independent variables, and car ownership behavior as the dependent variable, to quantify the impact of the selected built-environment attributes on car ownership behavior, and to explore the individual level. The nonlinear relationship between individual family attributes, neighborhood-scale and city-scale built-environment characteristics and car ownership, and the marginal effect of each variable of car ownership behavior, is analyzed and studied [40]. Compared with traditional statistical models, the GBDT model can not only quantify the degree of influence of each variable on car ownership behavior, but also obtain the potential nonlinear relationship between each variable and the dependent variable [41]. The gradient-boosting decision tree model has the advantages of strong interpretability, strong expansibility and robustness, and can deal with continuous variables and discrete variables at the same time. When the data sample is small, it can also obtain a good model-prediction effect.

The GBDT model is an algorithm to classify or regress the data set by using the additive model to reduce the residuals generated in the training process. The model can flexibly process data including continuous and discrete values, and enhance the robustness of the model to outliers by using appropriate loss functions. The specific algorithm flow is as follows.

(1) Initialize the weak learner, as shown in Formula (3). Let $\{(x_1, y_1), (x_2, y_2), \ldots, (x_N, y_N)\}$ be the set of independent variables and dependent variables, respectively. In this paper, the social and economic attributes of the individual level and the built-environment attributes at neighborhood scale and city scale are selected as independent variables, and the car ownership behavior of residents is the dependent variable.

$$F_0(x) = \text{argmin}_c \sum_{i=1}^{N} L(y_i, c) \tag{3}$$

where $c$ is the constant value that minimizes $F_0(x)$.

(2) Establish M classification regression trees, $m = 1, 2, 3, \ldots, M$:

(a) For $i = 1, 2, 3, \ldots, N$, compute the negative gradient corresponding to the $m$-th tree, the pseudo residual:

$$r_{m,i} = -\left[\frac{\partial L(y_i, F(x_i))}{\partial F(x_i)}\right]_{F(x) = F_{m-1}(x)} \tag{4}$$

(b) Using the CART regression tree to fit the data $(x_i, r_{m,i})$, the $m$-th regression tree with the leaf node region $R_{m,j}$ is obtained, $j = 1, 2, 3, \ldots, J_m$. Where $J_m$ represents the number of leaf nodes of the $m$-th regression tree.

(c) For $j = 1, 2, 3, \ldots, J_m$, calculate the best fitting value of $J_m$ leaf node regions:

$$c_{m,j} = \text{argmin}_c \sum_{x_i \in R_{m,j}} L(y_i, F_{m-1}(x_i) + c) \tag{5}$$

(d) Update the strong learner $F_m(x)$:

$$F_m(x) = F_{m-1}(x) + \sum_{j=1}^{J_m} c_{m,j} I(x \in R_{m,j}) \tag{6}$$

(3) The expression of strong learner $F_m(x)$ is obtained:

$$F_M(x) = F_0(x) + \sum_{m=1}^{M} \sum_{j=1}^{J_m} c_{m,j} I(x \in R_{m,j}) \tag{7}$$

(4) Calculate the mean value of the output of all decision trees, which can be used to quantify the impact of each variable on residents' car ownership behavior, as shown in Formula (8).

$$\begin{cases} I_k^2 = \frac{1}{N} \sum\limits_{n=1}^{N} I_k^2(T_n) \\ I_k^2(T_n) = \sum\limits_{j=1}^{J-1} \hat{\tau}_j^2 I(v(j) = k) \end{cases} \tag{8}$$

where $I_k^2$ is the effect of independent variable $x_k$ on residents' car ownership; $I_k^2(T_n)$ is the influence of independent variable $x_k$ on dependent variable in decision tree $n$; $\hat{\tau}_j^2$ is the adjustment factor; $j$ is the terminal node of the decision tree.

At the same time, the GBDT model can also generate partial correlation diagrams between independent and dependent variables to analyze the potential nonlinear relationship between them. The calculation method of the partial correlation between the independent variable and the dependent variable in the model is shown in formula (9).

$$\begin{cases} \overline{F}_s(x_s) = \frac{1}{V} \sum\limits_{v=1}^{V} [F(x_s, x_{ic})] \\ F_s(x_s) = E_{x_c}[F(x_s, x_c)] \end{cases} \tag{9}$$

where $x_s$ is the target independent variable; $x_c$ is an independent variable other than the target independent variable.

## 4. Results

In order to verify the applicability and performance of the GBDT model, this study selected Accuracy, AUC and F1 as indicators to evaluate the model. At the same time, this paper constructed a logit model and random forest model. The estimation results show that the three indexes of the GBDT model (0.8963, 0.9319, 0.8782) are better than the logit model (0.7729, 0.8539, 0.8058) and random forest model (0.8465, 0.9293, 0.8716).

### 4.1. The Impact of the Built Environment on Car Ownership Behavior

The individual socio-economic attributes of residents and the built-environment characteristics at neighborhood and city scale are modeled. Table 3 is the result of the influence of each variable on the car ownership behavior of residents based on the GBDT model. The model sets the sum of the degree of influence of all independent variables to 100%, so as to compare the degree of influence of different independent variables [42]. It can be seen from the model estimation results that the built environment at all scales and the individual family attributes of residents can affect car ownership behavior.

**Table 3.** Relative importance and ranking of the explanatory variables.

| Variable | Rank | Relative Contribution (%) |
|---|---|---|
| Individual level—personal family attributes | | |
| Gender | 14 | 0.45 |
| Age | 5 | 7.86 |
| Education | 11 | 3.63 |
| Family size | 7 | 6.62 |
| Number of members under 16 years old | 12 | 3.46 |
| Income(yearly) | 1 | 24.28 |
| Sum of relative contribution | | 46.3 |
| Neighborhood scale—built environment characteristics | | |
| Degree of mixed land use | 8 | 6.51 |
| Distance to bus station | 4 | 8.32 |
| Distance to city center | 3 | 9.47 |
| Neighborhood population density | 2 | 9.64 |
| Sum of relative contribution | | 33.94 |
| City scale—built environment characteristics | | |
| City population density | 9 | 6.20 |
| Whether there is a subway | 13 | 0.73 |
| The number of buses per ten thousand people | 6 | 7.18 |
| Road area per capita | 10 | 5.65 |
| Sum of relative contribution | | 19.76 |

The model results show that among the three types of influencing factors, the individual family attributes at the individual level lead to the highest proportion of car ownership, reaching 46.3%, which is consistent with the conclusion of Wang [39] that the most important factor affecting residents' car ownership behavior is individual socio-economic attributes. Specifically, the impact of household income accounts for up to 24.28%, which is the most influential attribute among all variables. This is consistent with the conclusion of Jiang [39] in the study of travel data in Jinan that household income has the greatest impact on car ownership behavior.

The degree of influence of the built-environment attributes at the neighborhood scale on car ownership is 33.94%, which indicates that the built-environment attributes at the neighborhood scale have a significant impact on car ownership, which is similar to the research conclusion of Ding [43], that is, the built-environment attributes of the neighborhood layer are crucial among the factors affecting car ownership. In addition, the impact of distance to the city center and neighborhood population density on car ownership is more than 9%, ranking second and third among all variables, second only to household income. In addition, the degree of influence of the built environment at the city scale is 19.76%. The number of buses per 10,000 people, urban population density and road area per capita have significant impacts on residents' car ownership, with the degree of influence of 7.18%, 6.20% and 5.65%, respectively.

*4.2. Nonlinear Effects of the Built Environment on Car Ownership*

The GBDT model can also model the nonlinear effects of various socio-economic attributes and different levels of built-environment attributes on car ownership.

Figure 2 shows the non-linear influence curve of household income on car ownership. As shown in Clark et al.'s research [44], household income has an impact on residents' car ownership behavior. It can be seen that the non-linear relationship between the two is significant. There is a positive correlation between family income and car ownership.

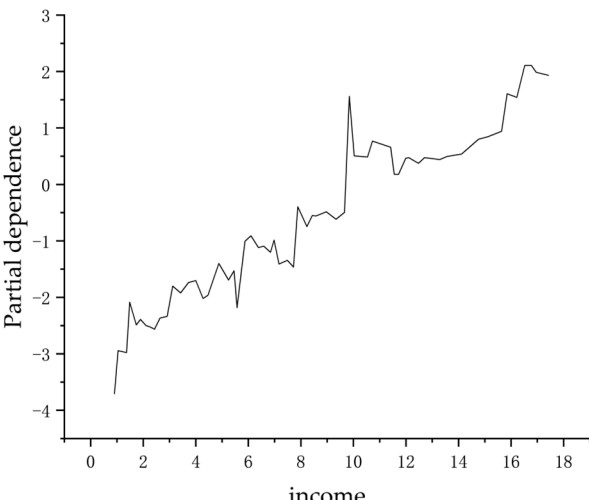

**Figure 2.** Nonlinear effects of income on car ownership.

There is also a positive correlation between family size and car ownership. According to Figure 3, when the family size is between 1–2 people, the probability of residents owning cars is low. When the family size is between 3–5 people, the probability of residents owning a car increases linearly; when the family size exceeds 5 people, the probability of residents owning a car fluctuates, but the overall stability is at a high level. Zhang et al. [45] also found that family structure has a significant impact on residents' car ownership. The reason for this situation may be that due to the increase in the family population, family travel is more dependent on cars, resulting in an increase in the probability of residents owning cars.

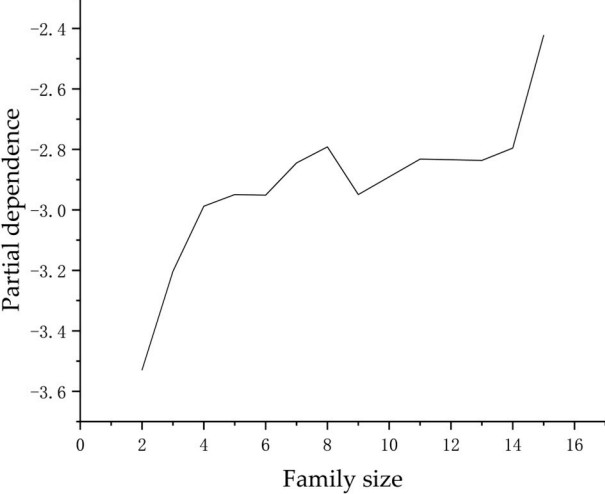

**Figure 3.** Nonlinear effects of family size on car ownership.

Figure 4 shows the nonlinear effect curve of the degree of mixed land use on car ownership. As is shown in the figure, the effect curve of mixed land use on residents' car ownership behavior shows a decreasing trend as a whole. When the degree of mixed land use of the community is in the range of 0–0.3, the probability of residents owning cars is higher. When the land use mix of the community is in the range of 0.3–0.6, the possibility of residents' car ownership is greatly reduced; when the land use mix of the community exceeds 0.6, the probability of residents owning cars fluctuates, but the overall trend tends to be stable and remains at a low level. Therefore, for areas with low levels of mixed land use, the increase in car ownership can be slowed down by appropriately enriching the living facilities of the community to increase the local mixed land use.

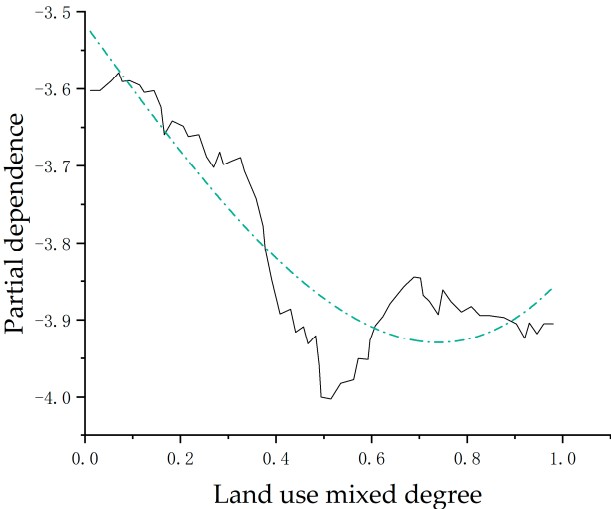

**Figure 4.** Nonlinear effects of degree of mixed land use on car ownership.

Figure 5 shows the nonlinear effect curve of distance to city center on car ownership. In the built-environment characteristics at the neighborhood scale, the distance from the residence to the city center has a significant threshold effect on the car ownership behavior of residents. When the distance from the residence to the city center is less than 5 km, the probability of car ownership decreases with the increase in the distance from the residence to the city center. When the distance from the residence to the city center is in the range of 5–11 km, the probability of residents owning a car increases linearly; when the distance exceeds 11 km, the probability of residents owning a car tends to be stable and remains at a high level. This is consistent with the conclusion of Sun et al. [46] in their study of Shanghai travel data which showed that people living farther away from the city center tend to drive more. Therefore, by rationally allocating buildings such as urban commercial centers or constructing a multi-center urban building layout, the probability of residents owning cars can be adjusted.

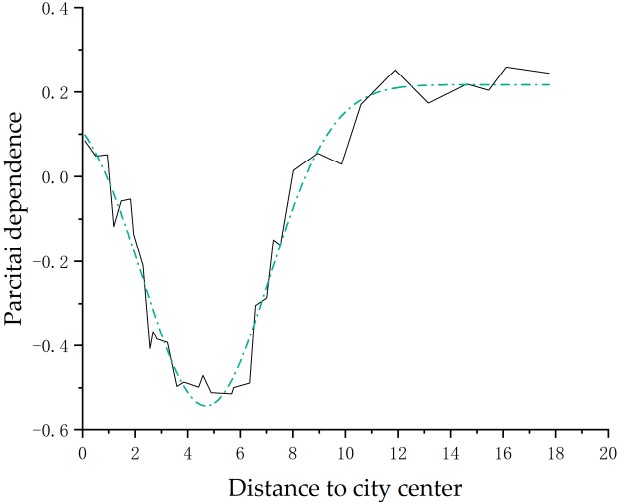

**Figure 5.** Nonlinear effects of distance to city center on car ownership.

In the built-environment characteristics at city scale, the number of buses per ten thousand people is the most important factor for car ownership, at 7.18%. From Figure 6, it can be seen that, although there are many noises in the relationship between this variable and car ownership behavior, as a whole, with the increase in the number of buses per 10,000 people, the probability of car ownership shows an increasing trend. The reason for this finding may be related to the year of data file selection. In the early stage of urban

construction, the increase in bus vehicles means that the level of urban road traffic is on the rise, which will promote residents' car ownership.

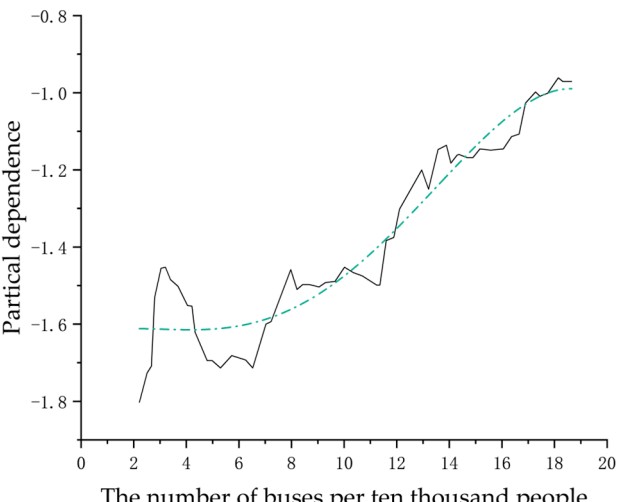

**Figure 6.** Nonlinear effects of the number of buses per ten thousand people on car ownership.

In terms of road area per capita, its impact on car ownership accounts for 5.65%, which is positively correlated with car ownership. It can be seen from Figure 7 that when the road area per capita of residents is less than 14 square meters, the probability of urban residents owning cars remains at a low level. When the road area per capita is 14–17 square meters, the probability of residents owning cars shows a linear upward trend. When the road area per capita is greater than 18 square meters, the probability of residents owning cars shows a steady growth trend. In general, the increase in urban per capita road area will promote residents' car ownership.

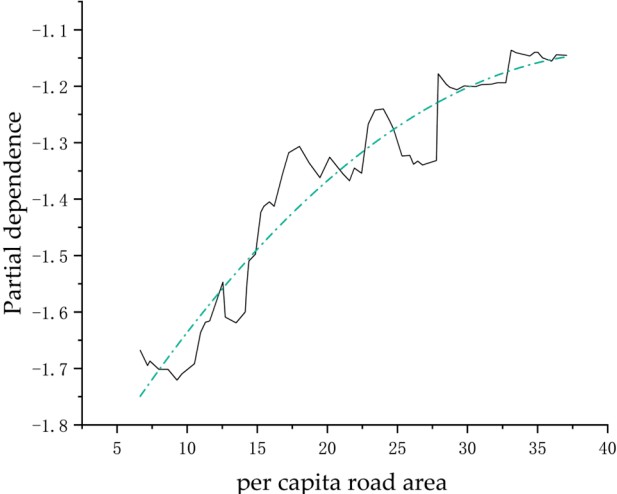

**Figure 7.** Nonlinear effects of per capita road area on car ownership.

## 5. Conclusions

(1) Individual family attributes have the greatest impact on car ownership behavior, accounting for 46.3%. Specifically, family size and household income are positively correlated with residents' car ownership behavior.

(2) The impact of the built-environment attributes at the neighborhood scale accounts for 33.94%, just next to the personal economic attributes of the residents, which indicates that car ownership behavior is closely related to the current degree of mixed land use of the neighborhood, the distance to the bus station and the city center, and the neighborhood

population density. Among these, the distance to the city center and car ownership are generally positively correlated, and the probability of residents owning cars is the lowest when the distance between the residence and the city center is 10–13 km.

(3) The impact of the urban built environment on car ownership behavior is the smallest, but the share is 19.76%. It can be seen that adjusting the urban built environment is also conducive to reducing motorized travel, thereby slowing down the growth rate of urban car ownership. In particular, car ownership by residents can be reduced by optimizing the urban public transport system and road layout.

Finally, the impact of non-motorized vehicles on residents' car ownership behavior is also important [47]. Due to the limitations of the data, our data cover many cities, and it is difficult to obtain the shared bicycle data of all sample cities. Therefore, there is no further analysis of the impact of non-motorized travel. With the wide application of big data in micro-mobility [48], the mechanism of its impact on residents' car ownership can be further explored.

## 6. Discussion

In contrast to other attributes, the built environment, as an external environment built for human activities, is still adjustable in urban planning and construction. Exploring the relationship between the built environment and residents' car ownership behavior has a high guiding significance for urban planning and policy formulation. Our study found that urban land use planning should try to limit the residential area and the downtown area to between 2–8 km, which can reduce the probability of residents having cars; for areas with low land use mix, residents' cross-regional travel can be reduced by appropriately enriching the living facilities around the community, thereby reducing residents' car ownership.

**Author Contributions:** Conceptualization, L.W. and C.Y.; Methodology, L.W., X.Y. and C.Y.; Software, X.Y. and H.O.; Resources, Ming Yang; Writing—original draft, X.Y.; Writing—review & editing, L.W., C.Y., M.Y. and H.O. All authors have read and agreed to the published version of the manuscript.

**Funding:** This research was funded by The National Natural Science Foundation of China (NO. 72204114), The Graduate Student Scientific Research Innovation Projects in Jiangsu Province (NO. SJCX20_0278).

**Institutional Review Board Statement:** Not applicable.

**Informed Consent Statement:** Not applicable.

**Data Availability Statement:** Not applicable.

**Conflicts of Interest:** The authors declare no conflict of interest.

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
