# Peer review of "Car Ownership Behavior Model Considering Nonlinear Impacts of Multi-Scale Built Environment Characteristics"

_sustainability, doi:10.3390/su15129658_

Round 1

Reviewer 1 Report

The research paper focuses on the impact of the built environment on residents' car ownership behavior. The paper explores the factors influencing residents' car ownership and proposes urban traffic planning and policy measures to promote green travel consciousness and reduce car dependence. While previous research has mainly focused on the linear relationship between built environment characteristics and car ownership, the authors argue that further studies have revealed the existence of nonlinear relationships. Therefore, they propose the use of machine learning models (GBDT) to explore the nonlinear relationship between different scales of the built environment and car ownership behavior. In terms of research data, the authors use comprehensive survey data and also gather data from statistical yearbooks of various cities. The data cover individual family attributes, neighborhood-scale built environment characteristics (such as land use mix and accessibility), and city-scale built environment attributes (including urban population density, presence of subway systems, and road area per capita).

A major problem this research suffers from is a lack of analysis of the effects of non-motorised transport, particularly shared micromobility (e.g. shared bicycles) that has become very popular in China over the past decade. Micromobility offers a viable solution to the last-mile problem an makes car alternatives attractive. It also offers a lot of data for analysis that could be used in this research. E.g. see

Mangold, M., Zhao, P., Haitao, H., & Mansourian, A. (2022). Geo-fence planning for dockless bike-sharing systems: a GIS-based multi-criteria decision analysis framework. Urban Informatics, 1(1), 17.

Also, the analysis is very basic. The authors need to elaborate on the discussion and clarify the significance of the results.

There are Inconsistent Verb Tenses e.g. "Wang et al. [38] modeled residents' car ownership behavior from two levels of workplace and residence, and found that the built environment attributes of both places have a significant impact on residents' car ownership behavior."

Author Response

We wish to thank the reviewer for his/her helpful comments. We have copied-and-pasted these below and after each comment we have inserted our response. All newly

added/revised texts to the manuscript are highlighted in blue.

Comment 1:

The research paper focuses on the impact of the built environment on residents' car ownership behavior. The paper explores the factors influencing residents' car ownership and proposes urban traffic planning and policy measures to promote green travel consciousness and reduce car dependence. While previous research has mainly focused on the linear relationship between built environment characteristics and car ownership, the authors argue that further studies have revealed the existence of nonlinear relationships. Therefore, they propose the use of machine learning models (GBDT) to explore the nonlinear relationship between different scales of the built environment and car ownership behavior. In terms of research data, the authors use comprehensive survey data and also gather data from statistical yearbooks of various cities. The data cover individual family attributes, neighborhood-scale built environment characteristics (such as land use mix and accessibility), and city-scale built environment attributes (including urban population density, presence of subway systems, and road area per capita).

A major problem this research suffers from is a lack of analysis of the effects of non-motorised transport, particularly shared micromobility (e.g. shared bicycles) that has become very popular in China over the past decade. Micromobility offers a viable solution to the last-mile problem an makes car alternatives attractive. It also offers a lot of data for analysis that could be used in this research. E.g. see

Mangold, M., Zhao, P., Haitao, H., & Mansourian, A. (2022). Geo-fence planning for dockless bike-sharing systems: a GIS-based multi-criteria decision analysis framework. Urban Informatics, 1(1), 17.

Responses1:

Thank you for your suggestion about the effects of non-motorised transport. This issue raised by the reviewer is very important, we have added the following literature review after the conclusion that was missing from the original manuscript. Please find the revised portion below, and we have also made the necessary modifications to the original manuscript, see page 12. The micromobility still needs additional relevant studies, we have made the following additions: due to the limitations of the data, we cover a lot of cities, and it is difficult to obtain the shared bicycle data of all sample cities. Therefore, there is no further analysis of the impact of non-motorized travel. However, this question is very important. We explain it in the research defect section of the article and quote the literature you suggest.

Finally, the impact of non-motorized vehicles on residents' car ownership behavior is also important [48]. Due to the limitations of the data, our data cover many cities, and it is difficult to obtain the shared bicycle data of all sample cities. Therefore, there is no further analysis of the impact of non-motorized travel.

Comment 2:

Also, the analysis is very basic. The authors need to elaborate on the discussion and clarify the significance of the results.

Responses2:

Thank you for your suggestion,We add some comparisons with existing research in 4.2.

Firstly, on the analysis of the conclusion of family size on residents ' car ownership behavior, we add the following description in blue.

There is also a positive correlation between family size and car ownership behavior. According to Figure 3, when the family size is between 1-2 people, the probability of residents owning cars is low. When the family size is between 3-5 people, the prob-ability of residents owning a car increases linearly; when the family size exceeds 5 people, the probability of residents owning a car fluctuates, but the overall stability is at a high level. Zhang et al. [46] also found that family structure has a significant im-pact on residents ' car ownership behavior. The reason for this situation may be that due to the increase of family population, family travel is more dependent on cars, resulting in an increase in the probability of residents owning cars.

Secondly, in the conclusion analysis of the distance to the city center on the car ownership behavior of residents, we add the following description in blue.

Figure 4 shows the nonlinear effect curve of the built environment of the neighborhood scale on the residents' cars. In the built environment characteristics of the neighborhood scale, the distance from the residence to the city center has a significant threshold effect on the car ownership behavior of residents. When the distance from the residence to the city center is less than 5 kilometers, the probability of car owner-ship decreases with the increase of the distance from the residence to the city center. When the distance from the residence to the city center is in the range of 5-11 km, the probability of residents owning a car increases linearly; when the distance exceeds 11 kilometers, the probability of residents owning a car tends to be stable and remains at a high level. This is consistent with the conclusion of Sun [47] in their study of Shanghai travel data that people living farther away from the city center tend to drive more. Therefore, by rationally allocating buildings such as urban commercial centers or constructing a multi-center urban building layout, the probability of residents owning cars can be adjusted.

Thirdly, for the conclusion analysis of the number of buses per 10,000 people on the car ownership behavior of residents, we add the following description in blue.

In the built environment characteristics of the city scale, the number of buses per ten thousand people is the most important for car ownership behavior, which is 7.18 %. From Figure 5, it can be seen that although there are many noises in the relationship between this variable and car ownership behavior, as a whole, with the in-crease of the number of buses per 10,000 people, the probability of car ownership shows an increasing trend. The reason for this finding may be related to the year of data file selection. In the early stage of urban construction, the increase of bus vehicles means that the level of urban road traffic is on the rise, which will promote residents' car ownership behavior.

Finally, in order to better demonstrate the results of this paper, we increase the influence of land use mix on residents' car ownership behavior. (In page 10)

Figure 4 shows the nonlinear effect curve of land use mixed degree on car ownership. As is shown in the figure, the effect curve of land use mixing on residents' car ownership behavior shows a decreasing trend as a whole. When the land use mixing degree of the community is in the range of 0-0.3, the probability of residents owning cars is higher. When the land use mix of the community is in the range of 0.3-0.6, the possibility of residents' car ownership behavior is greatly reduced; when the land use mix of the community exceeds 0.6, the probability of residents owning cars fluctuates, but the overall trend tends to be stable and remains at a low level. Therefore, for areas with low land use mix, the increase in car ownership can be slowed down by appropriately enriching the living facilities of the community to increase the local land use mix.

Figure 4. Nonlinear effects of land use mixed degree on car ownership.

Reviewer 2 Report

This is an interesting paper. There are a few things the authors should either clarify or strengthen. The first is their discussion of the limitations of prior statistical studies. As written, they emphasize non-linear relationships among variables, but they don't explain why other statistical methods can't address these or alternately why their approach is better. Second, the authors should add some text explaining why they selected the variables they included in their model, and they should define the spatial units over which variables such as density measures were defined. Third, they should more clearly explain some of the statistical findings. For example, they report a positive relationship between the bus variable and car ownership. This seems to be a counterintuitive finding, and it would be helpful for the reader for the authors to clarify why this (and the other statistical results) might occur. Fourth, the authors might consider whether the literature review would be strengthened with an explicit framing at the start that it is organized by the methodologies used instead of the variables, which is more traditional. Without this framing, it is easier for the reader to get confused about the themes the authors seek to emphasize.   Finally, the authors should consider strengthening the conclusion to emphasize the broader importance of the study and its implications either for scholarship or practice. 

Generally clear. Minor proofreading suggested.

Reviewer 3 Report

The subject of the article is interesting, and the problem is worth exploring.
It would be necessary to characterize the authors' barriers and limitations in their experiments.

Supplementing the paper with an additional "Discussion of Results" section would be reasonable. In the discussion of the results, it would be appropriate to refer to other studies in this area and point out the differences, advantages and disadvantages of the solutions presented by the authors and indicate the contribution to scientific research.
The purpose of conducting the research and research methods is entirely
clear, but needs clarification:

- In Figures 3 to 5, the authors of the article show with a dashed line the dependency models of the selected variables. Can the models be described as functions and their fit verified?

Reviewer 4 Report

I found the paper to be well-written with a good discussion of the background of the model they were trying to build and other similar research that has contributed to the field.  Generally, the paper flowed well and it was easy to understand the model and the principal findings from the model.  I did not see any major issues but saw one minor grammatical/typo error.  On page 2, line 59 the authors should remove the words "most studies" after "many studies".  

Author Response

We wish to thank the reviewer for his/her helpful comments. We have copied-and-pasted these below and after each comment we have inserted our response. All newly added/revised texts to the manuscript are highlighted in blue.

Comment 1:

I found the paper to be well-written with a good discussion of the background of the model they were trying to build and other similar research that has contributed to the field. Generally, the paper flowed well and it was easy to understand the model and the principal findings from the model.  I did not see any major issues but saw one minor grammatical/typo error. On page 2, line 59 the authors should remove the words "most studies" after "many studies".

Responses1:

Thanks for your careful checks. We are sorry for our carelessness. Based on your comments, we have made the corrections to make the word harmonized within the whole manuscript. We have corrected L59 page 2in the manuscript. “Many studies have focused on the impact of CBD distance on car ownership.” Corresponding modifications were also made in the original manuscript.

Round 2

Reviewer 1 Report

The paper is improved. Most of my comments are addressed. I suggest the authors cite this latest paper in the final paragraph on how emerging big data could help overcome your mentioned data issues when analysing the impact of non-motorised transport.

Schumann, H., Haitao, H., & Quddus, M. (2023). Passively generated big data for micro-mobility: State-of-the-art and future research directions. Transportation Research Part D: Transport and Environment, 121103795.

N.A.

Author Response

Authors’ responses to reviewers’ comments on the manuscript:

Manuscript ID: sustainability-2344598

Title: Car Ownership Behavior Model Considering Nonlinear Impacts of Multi-scale Built Environment Characteristics

Dear Editor Ms. Corrine Chen and Reviewers,

We sincerely thank the editor and all reviewers for their valuable feedback that we have used to improve the quality of our manuscript. If there are any other modifications we could make, we would like very much to modify them and we really appreciate your help.

Best regards,

Xiaorui Yuan

[Response to reviewer]:

We wish to thank the reviewer for his/her helpful comments. We have copied-and-pasted these below and after each comment we have inserted our response. All newly

added/revised texts to the manuscript are highlighted in blue.

Comment 1:

The paper is improved. Most of my comments are addressed. I suggest the authors cite this latest paper in the final paragraph on how emerging big data could help overcome your mentioned data issues when analysing the impact of non-motorised transport.

Schumann, H., Haitao, H., & Quddus, M. (2023). Passively generated big data for micro-mobility: State-of-the-art and future research directions. Transportation Research Part D: Transport and Environment, 121, 103795.

Responses1:

We sincerely appreciate the valuable comments. We read the literature recommended by reviewer in detail, understood the research status of micro-mobility, realized the necessity of micro-mobility and big data for the follow-up research of this paper. We have checked the literature carefully and added the reference on micro-mobility and big data in our manuscript.

Finally, the impact of non-motorized vehicles on residents' car ownership behavior is also important [48]. Due to the limitations of the data, our data cover many cities, and it is difficult to obtain the shared bicycle data of all sample cities. Therefore, there is no further analysis of the impact of non-motorized travel. With the wide application of big data in micro-mobility [49], the mechanism of its impact on residents' car ownership behavior can be further explored.

Reviewer 3 Report

I don't have any more comments.

Author Response

We really appreciate your concern and support for our research.